# Efficacy of a Comprehensive and Personalised Approach for Frail Older People in Valencia (Spain): A Pre–Post Controlled Trial

**DOI:** 10.3390/healthcare12171754

**Published:** 2024-09-03

**Authors:** Mirian Fernández-Salido, Tamara Alhambra-Borrás, Jorge Garcés-Ferrer

**Affiliations:** Research Institute on Social Welfare Policy (POLIBIENESTAR), Universitat de València, 46022 Valencia, Spain; tamara.alhambra@uv.es (T.A.-B.); jordi.garces@uv.es (J.G.-F.)

**Keywords:** older people, frailty, value-based healthcare, personalised care, integrated care, pre–post controlled trial

## Abstract

Frailty is a common condition in older adults that negatively impacts health and quality of life. This study evaluated a comprehensive, personalised, and coordinated intervention under the value-based care approach to address frailty’s multidimensional nature in older people in the primary care setting. It employed a pre–post randomised controlled design involving 242 frail individuals aged over 65 years living in the community in Valencia (Spain) between 2021 and 2023. Assessments were conducted at baseline, 12 months (immediately post-intervention), and 18 months. The intervention included a personalised care plan supported by technology, with monthly motivational follow-ups and plan updates by health professionals and participants. Outcomes were measured using an assessment questionnaire that included the International Consortium for Health Outcomes Measurement dataset for the older population: physical health, physical functioning, general mental health, satisfaction with social activities and relationships, ability to carry out usual social roles and activities, pain, general quality of life, loneliness, physical frailty, psychological frailty, and social frailty. The study found significant improvements in physical frailty, quality of life, reduced health resource use and hospitalisations and lower levels of pain and depression/anxiety compared to baseline. The findings suggest further research into value-based care approaches, emphasizing the development and activation of personalised, comprehensive programs for older individuals with frailty.

## 1. Introduction

Population ageing continues to boom with an accelerating expansion of the older population worldwide [1,2]. According to United Nations projections, it is estimated that in 2025, worldwide, one in six people will be 65 years of age or older [3], which represents approximately 31% of the European population in 2100 [4]. This progressive and accelerated increase in population ageing brings with it multiple challenges for health systems and public health budgets, as it is associated with a sharp increase in care services [5]. Specifically, the increasing frailty, disability, and morbidity associated with ageing impose a growing burden on health systems that require reform to meet the growing need for medical and social resources [6,7]. Identifying effective and efficient interventions in terms of promoting the health-related quality of life of older people is a European policy priority, and clinicians, policymakers, healthcare managers, scientists, and researchers should consider the potential of comprehensive and coordinated approaches to care [5,8].

Among the different conditions faced by the older population, frailty is one of the most pressing, yet preventable and reversible. The determinants of frailty are driven by multiple interrelated risk factors that can be reversed and prevented, such as falls, depression, unhealthy diet, physical inactivity, and social isolation [9,10,11]. Frailty is a controversial term that encompasses multiple meanings depending on the context and is interpreted differently by clinical and non-clinical settings. While for the older population that suffers from frailty, frailty may be associated with negative connotations because of its association with physical deterioration, generating rejection in older people because of the stigma of bodily deterioration associated with old age, for those who label it, frailty also encompasses multiple conceptualisations despite the consensus that it is a clinical category that needs to be detected and addressed to reduce its negative impact on the health of the population [12,13]. Although frailty is common among older adults, epidemiological data and interventions aimed at preventing and reversing this condition have traditionally identified and managed it based on the phenotypic model, which emphasizes physical frailty [14]. Extensive literature highlights the necessity of identifying and addressing frailty through a multidimensional, integrated, and holistic approach to care. This is essential because frailty impacts multiple aspects of health, including physical, psychological, cognitive, social, and emotional domains [15,16]. In this way, the recent multidimensional conceptual model understands frailty as the lack of harmonious interaction between multiple dimensions leading to homeostatic instability [17]. From this holistic approach, frailty is therefore understood as a dynamic state resulting from deficits in any of the social, psychological, and physical domains that contribute to health, and therefore requires the identification, assessment, and care of the condition from a biopsychosocial approach [18,19].

Achieving better health outcomes for patients requires reorganising care for patients efficiently and effectively, and in this regard, health systems are advocating for innovative models of care that move away from a volume-based approach to care towards a value-based approach to care, supported by digital solutions [20]. The goal of this approach is the delivery of value to patients with the understanding that value is defined as improved health outcomes achieved from the entire care process [21]. In this context, the quality of care is multidimensional, just like the life of an older person experiencing frailty [22]. This necessitates assessment measures that cover all aspects of human functioning—physical, psychological, and social—related to frailty, highlighting the importance of a holistic approach. The evidence confirms that the Tilburg Frailty Instrument (TFI) is a key self-report instrument for assessing frailty in older people living in the community [23]. Furthermore, taking a holistic approach to addressing frailty in community-dwelling older adults can help manage other issues such as social isolation and loneliness. Evidence indicates that socially isolated older adults are much more likely to develop both physical and psychological frailty, as loneliness often leads to emotional depression [24].

Given the positive and significant relationship between loneliness and frailty, where each can impact the other, studies recommend not only a holistic assessment that addresses loneliness but also multidimensional and personalised interventions that focus on enhancing psychosocial resources [25]. In this regard, recent literature highlights the need to raise awareness among older adults about actively investing in their existing social ties, such as family and friends. It also highlights the necessity of enhancing their motivation to actively improve their situation [26]. Following this line, motivational interviewing (MI) can be an effective therapeutic technique for its ability to promote communication and commitment, encouraging changes toward healthy lifestyles. This approach is empathic and collaborative, with achievable goals set according to the needs and interests of the person being interviewed [27]. Given that evidence confirms a positive association between information and communication technology (ICT) use and social support, incorporating ICT into interventions to address psychosocial frailty can be beneficial for maintaining and extending social connections [28,29].

However, studies on value-based care programmes supported by technological solutions that address the multidimensional nature of frailty and evaluate their effects on older patients with frailty are currently scarce and have limitations, such as the lack of controlled trials [22]. In this study, the purpose is to analyse the effect of a comprehensive and personalised approach based on motivational interviewing, supported by a digital tool on the reduction of frailty, especially psychosocial frailty, in older adults.

This study is part of the ValueCare project—Value-based methodology for integrated care supported by ICT—an initiative funded by the European Commission under the Horizon 2020 programme. The project brings together a consortium of 17 partners from eight European countries. The aim of the ValueCare project is to provide efficient and outcome-based integrated care (both health and social) for people aged 65+ with frailty, cognitive impairment, and/or multiple chronic conditions. The ValueCare intervention has been implemented and validated in study sites located in seven European countries: Valencia in Spain, Rijeka in Croatia, Athens in Greece, Cork/Kerry in Ireland, Coimbra in Portugal, and Rotterdam in the Netherlands. In each of these sites, the ValueCare intervention focuses on addressing a specific health condition. Through value-based methodologies supported by digital solutions, the project aims to improve the quality of life of this population. In Spain (Valencia), the ValueCare approach supported by a digital tool and therapeutic tool based on motivational interviewing has been implemented for people over 65 diagnosed with frailty.

This study aims to increase knowledge about the potential effects of a comprehensive, personalised, and coordinated intervention, supported by digital solutions, in the context of frailty. The main objective of this study was to evaluate the effects of ValueCare intervention on frailty, including its social, psychological, and physical domains. Additionally, this study analysed the impact of the intervention on other health-related variables, namely, global health, loneliness, health-related quality of life, and the use of health resources, the hypothesis being that after intervention, participants would also show an improvement in these variables.

## 2. Materials and Methods

### 2.1. Study Design

This study was a randomised, parallel-controlled clinical trial with a control group and an intervention group and a pre- and post-evaluation design. It was conducted with the support of the Malvarrosa Clinic Health Department of the city of Valencia (Spain). Participants included in this study were assessed at baseline and at 12 months after completion of the intervention. All study participants gave written informed consent. The clinical trial protocol had previously been approved by the Human Research Ethics Committee (HREC) of the Experimental Research Ethics Committee of the University of Valencia (7 May 2020), and it has been registered in the International Standard Randomised Controlled Trial registry (ISRCTN25089186; registration date 16 November 2021). A full description of the methods, design, and procedure is available in the trial protocol [30].

### 2.2. Study Participants: Inclusion and Exclusion Criteria

Recruitment for this study was carried out in the city of Valencia (Spain) with the support of a total of seven primary care health centres belonging to the Malvarrosa Clinic Health Department. Citizens aged 65 years and over who lived in the community were telephoned by the socio-health professionals of each of the participating health centres for recruitment according to the clinical history and the established inclusion criteria. Persons eligible to participate in the current study were those who met the following eligibility criteria: (a) aged 65 years or older, (b) frailty, and (c) independent residence in the community and affiliation to one of the seven participating health centres belonging to the Malvarrosa Clinic Health Department. In addition, citizens were ineligible to participate in the study when they presented any of the following conditions: (a) cognitive impairment, (b) significant dependency, (c) institutionalisation, (d) inability to make an informed decision regarding participation in the study, and (e) lack of Spanish language proficiency. Participants were randomly assigned to intervention and control groups. Randomisation was performed using a computer-generated list of random numbers through the Oxford Minimisation and Randomisation (OxMaR) system to ensure concealment of the randomisation sequence. Of the 242 who were eligible for randomisation, 122 were assigned to the intervention group (to receive the 12-month ValueCare programme) and 120 to the control group (to receive no intervention/receive usual care). Adherence to the study was estimated to be 71.9% (individuals who completed the intervention against those lost in follow-up).

### 2.3. Measures

#### 2.3.1. Primary Outcome Variables

Frailty was assessed through the Tilburg Frailty Index (TFI) [31]. The TFI is based on a holistic approach to frailty, including physical, psychological, and social domains. It is composed of three subscales: physical frailty, psychological frailty, and social frailty, as well as a total score on frailty. The TFI’s total score ranges from 0 to 15, with a score ranging from 0 to 8 for physical frailty, 0 to 4 for psychological frailty, and 0 to 4 for social frailty. Higher scores refer to greater frailty, and scores greater than or equal to 5 indicate the presence of frailty. TFI has shown robust evidence of reliability and validity [22].

#### 2.3.2. Secondary Outcome Variables

Global health was assessed using the PROMIS-10 Global Health survey which includes two subscales on physical health and mental health. Raw PROMIS-10 scores were converted to standardised *t*-score values [32]. A *t*-score of 50 represents the mean of the general population, and higher scores indicate better physical and mental health [32] according to the established *t*-score cut-offs for fair-to-poor health ratings (physical health < 42 and mental health < 40) [33].

Loneliness was assessed using the UCLA Three-Item Loneliness Scale [34]. This scale asks how often a person felt that they (1) lacked companionship; (2) were left out; and (3) were isolated from others on a 3-point Likert scale coded from 1 (hardly ever) to 3 (often). The scores for each individual question can be added together to give a possible range of scores from 3 to 9, with higher scores indicating greater loneliness. People who score 3–5 are identified as ‘not lonely’ and people scoring 6–9 as ‘lonely’. The Spanish version of the scale has shown good psychometric properties [35].

Health-related quality of life was evaluated with the 5Q-5D-5L [36]. The EQ-5D instrument was developed by EuroQol (www.euroqol.org—accessed on 14 February 2020) and is one of the most reliable generic health-related quality of life measurement and widely used around the world. The EQ-5D-5L is divided into five dimensions—mobility, selfcare, usual activities, pain/discomfort, anxiety/depression—within five levels of problem severity in responses from ‘no problems’ to ‘extreme problems’. The measure includes a visual analogue rating scale (VAS), which was not included in our current study.

Use of healthcare resources was evaluated by asking participants for their number of primary care visits and hospitalisation days over the past 12 months (at baseline and at post-intervention).

### 2.4. Intervention

The ValueCare intervention to improve psychosocial frailty and loneliness in older people by encouraging behavioural change towards healthy lifestyles and greater social engagement lasted 12 months and consisted of three elements: (i) periodic motivational social prescription sessions under the motivational interviewing approach (ii) support through the implementation of monthly social workshops in the participating primary care centres, and (iii) monitoring and support of the intervention through the ValueCare digital solution.

Motivational sessions developed using the approach of motivational interviewing were offered monthly during the 12 months of intervention, with a duration of 30 min–1.5 h. A protocol was established to guide the sessions, its development was supported by scientific evidence on the effectiveness of this methodology to promote behavioural change, commitment to change, and adherence to the intervention. The sessions focused on identifying the individual’s psychosocial needs, assessing their readiness for change, and stimulating motivation to prepare them for change. They also helped participants explore and resolve ambivalences related to unhealthy behaviours or habits. Ultimately, the sessions involved jointly setting objectives within the framework of social prescription through a negotiation process between the professional and the patient to facilitate the transition toward healthy habits. The motivational sessions were conducted by social and health professionals such as social workers and psychologists with knowledge of the clinical condition of each participant. The objectives established in terms of social prescription allowed the professional to present to the participant the existing community resources that could favour their health and well-being, as well as the social workshops to be implemented within the framework of the intervention. During the development of the motivational sessions, an atmosphere of trust was established based on empathy through active and reflective listening in which participants were enabled to explore and resolve ambivalence in changing behaviour towards healthy lifestyles, promoting intrinsic motivation that encourages change.

Social participation in social workshops was used to support the achievement of the social prescription goals set in the motivational sessions in terms of reducing loneliness and improving the psychosocial frailty of older people. Socio-health professionals organised a monthly group social workshop in each primary care centre, where participants were invited to attend with the aim of increasing social connections and networks with other project participants. The workshops covered different themes: (a) healthy ageing workshop on the importance of physical activity and social prescription, (b) workshop on healthy nutrition based on the knowledge of the Harvard plate for a balanced diet, (c) workshop on the use and usefulness of new technologies in the framework of the ValueCare digital solution, (d) workshop on art and cognitive stimulation to exercise functions such as memory, (e) workshop on emotional management through plastic arts to work on the areas of expressiveness and emotional management, and (f) workshop on age discrimination and mistreatment of the elderly to work on tools to deal with ageism. Social workshops with a leisure character are beneficial in reducing loneliness and depression levels among their users, as well as favouring their life satisfaction and providing a sense of place, enjoyment, and support among older people that is rooted in a sense of relevance [37].

In addition, for older people, the use of digital technologies can be challenging, as they are generally later adopters of technological innovation [38]. In this sense, motivational interviewing together with social workshops were further employed as a way to ensure adherence and follow-up of the element. In this sense, participants were provided with a space where they were encouraged in each motivational session and each workshop to report any doubts or comments about the ValueCare digital application.

Both the goals to be achieved by older people agreed to in the motivational sessions and the description of and invitation to the social workshops were included in the ValueCare digital solution. This ValueCare app1 presented the personalised care plan for each of the participants in the intervention group, which was set up and monitored by the social and health professionals. In addition, the application had a messaging portal through which the professionals reminded the participants of the objectives to be achieved agreed to in the motivational sessions. In this way, the participants had access to their personalised care plan, in which they could also interact through the application thanks to the existence of a virtual coach who acted as a persuasive chatbot. This chatbot was based on dialogue, motivating the participants to achieve the prescribed objectives and reinforcing positive behaviours by confirming that the objective had been met or encouraging the participant to reach it. In addition, the ValueCare digital solution featured a section of content in audio-visual and text format on services available in the community to promote an active and healthy lifestyle, suggestions for increasing social interaction, and information about physical frailty, social frailty, and loneliness, among other relevant topics. A tablet was provided free of charge to each participant, and training sessions were developed on a voluntary basis to ensure the proper installation of the digital solution and teaching on the use and usefulness of the solution. It is worth mentioning that the ValueCare app and the tablets were available for use on a voluntary and unmonitored basis for up to six months after the end of the intervention. During these six months, researchers continued to voluntarily send personalised motivational messages to participants in the intervention group that included physical and nutritional recommendations, as well as information about health promotion activities available at their health centres. Participants were also encouraged to download the app on their mobile phones to continue accessing the audio-visual content portal available on the app. Finally, all participants were invited to a final event after the end of the intervention, where they received a paper guide with health recommendations and upcoming health promotion events available at their health centres. The aim of these non-intervention actions was to reduce the possible feeling of dropout at the end of the study by ensuring a staggered exit.

### 2.5. Data Analysis

Analysis of the effectiveness of the intervention was conducted through intragroup analyses to determine the evolution of outcome measures in the comparison and the intervention group, and intergroup analyses to compare the two groups in two different moments: pre-intervention and 12 months later.

The impact of the intervention on the outcome measures—frailty, global health, loneliness, health-related quality of life, and use of healthcare resources—was assessed using Student’s *t* test. For variables of a categorical nature, chi-squares tests were performed for intergroup analyses, and intragroup differences were analysed using McNemar’s test. A threshold of *p* < 0.05 for statistical significance was used. Additionally, effect size was calculated using Cohen’s d (small effect = 0.2, medium effect = 0.5, large effect = 0.8). Data were analysed using IBM SPSS Statistics version 28.

## 3. Results

Participants had an average age of 72.9, ranging from 65 to 90 years old, and included both females (73.6%) and males (25.6%). Intervention group participants were slightly younger than comparison group participants (mean age 72.8 vs. 73.1), and the percentage of women was also slightly higher in the intervention group (75.4% vs. 71.7%).

Participants in this study showed medium levels of frailty according to the TFI, for which scores greater than or equal to 5 indicate the presence of frailty. Regarding the effect of the intervention on frailty, as presented in Table 1, participants in the intervention group showed a slight reduction in the psychological and social domains of frailty and a slight increase in the physical domain and in overall frailty. The comparison group did not experience any changes in the social or psychological domains and showed a slight reduction in physical and overall frailty. However, none of these changes in frailty was found to be statistically significant for either of the groups.

Global health results, which were measured using the PROMIS-10 Global Health survey, showed that participants in both groups had fair global physical and mental health. As presented in Table 2, physical health was significantly improved among intervention group participants (*p* = 0.029) with a small effect size (0.22), while the comparison group showed no statistically significant improvement on this variable. Regarding mental health, improvement was found only for the intervention group, but this was not statistically significant (*p* = 0.238).

In terms of loneliness, no differences in the feeling of loneliness were found between the groups at baseline and at follow-up, as shown in Table 3. The intervention did not show any statistically significant improvement for either group, and participants remained identified as ‘not lonely’, according to the UCLA scoring.

As shown in Table 4, changes in health-related quality of life were found in both groups when comparing baseline with follow-up measurements on 5Q-5D-L5 general score. Both intervention and comparison group participants reported improvements in their quality of life after 12 months, and these changes were statistically significant. A medium effect size (0.61) was found for the intervention group, while for the comparison group, the effect size was smaller (0.45). Apart from the improvements found for the general score on quality of life, significant improvements were found for the pain and anxiety/depression subscales of the 5Q-5D-L5. Pain issues were reduced by 34.4% (*p* = 0.000) among the older adults who attended the intervention, and anxiety/depression issues were reduced by 27.1% (*p* = 0.000). On the other hand, in the same time period, comparison group participants also had reduced pain issues by 30.9% (*p* = 0.000) and anxiety/depression issues by 18.3% (*p* = 0.009).

Finally, the effects of the intervention on the use of healthcare resources are presented in Table 5. The average number of primary care visits was significantly reduced among intervention group participants (*p* = 0.036). Prior to being included in the intervention, this group visited the primary care doctor an average of 4.2 times per year, while after the intervention, this average was reduced to 3.4 visits per year. Hospitalisation was also reduced by 9% (*p* = 0.049) among intervention group participants. No effect was found in the use of healthcare resources among the comparison group on visits to primary care or hospitalisation.

## 4. Discussion

The increasing frailty that accompanies the trend of population ageing is a major public health problem that represents a significant burden on the healthcare system, given the consequences of this condition on the overall health and health-related quality of life of older people [39,40]. Despite this, existing epidemiological data as well as interventions implemented over the years have focused on the physical frailty phenotype. Although in the last two decades, the scientific community has striven to reach a consensus definition of frailty assessment, there is currently no international standard definition of frailty [41,42]. Recently a large body of literature has suggested that the condition of frailty should be identified and addressed from a multidimensional approach, giving rise to a new conceptual model of frailty based on the loss of harmonious interaction between different domains, which gives importance to the social and psychological domains in addition to the physical domain [43]. Given the variability in the identification, assessment, and management of frailty, we are faced with the absence of an adequate evidence base on effective interventions to manage frailty [44].

To our knowledge, this is the first study to investigate the effectiveness of a value-based, holistic, and personalised approach combining motivational techniques and social prescription supported by digital solutions in frail older adults.

In fact, the main objective of this study was to evaluate the effects of the ValueCare intervention on frailty, including its social, psychological, and physical domains. The results of the statistical analyses on frailty showed that for the participants in the intervention group, there was a slight reduction in the social and psychological domains of frailty, while physical frailty was slightly increased; however, these reductions were not statistically significant. Although the absence of previous studies similar to this one hinders a comprehensive comparison of our results, other randomised clinical trials have confirmed the efficacy of multifactorial interventions. These interventions included social supports like home telecare to prevent or delay the progression of frailty [45] and participation in psychosocial programs featuring practical and group activities, which significantly reduced frailty and improved functional health [46].

Additionally, this study analysed the impact of the intervention on other health-related variables, namely, global health, loneliness, health-related quality of life, and the use of health resources, the hypothesis being that after intervention, participants would also show an improvement in these variables. The results indicated that regarding global health, participants in the intervention group experienced significant improvements in physical health, while no statistically significant changes were observed for mental health. Previous studies support the effectiveness of multifactorial interventions in enhancing functional status [47]. In terms of loneliness, the intervention did not result in statistically significant improvement for either group according to the UCLA score. Regarding health-related quality of life, the analysis found that the ValueCare intervention led to statistically significant improvements in the overall 5Q-5D-5L quality-of-life score after 12 months of implementation for both the comparison and control groups. Previous studies with programmes that included physical and nutritional interventions have shown significant improvements for physical frailty compared to social programmes [48], suggesting that multicomponent intervention may be the key to improving overall frailty.

In this context, despite the limited results, it is noteworthy that study protocols have been developed for multifactorial interventions similar to the present study. These protocols include the assessment of psychosocial frailty and the incorporation of motivational interviewing and social prescription to reduce frailty [49]. Significant improvements were also observed after the intervention period compared to baseline for the pain and anxiety/depression subscales, with a reduction in both pain and anxiety among participants in the intervention group. Other multifactorial studies have reported similar findings, showing that addressing anxiety and depression as elements related to frailty leads to improvements in these conditions following the implementation of a multicomponent intervention [50].

Finally, concerning the use of healthcare resources and hospitalisations, the intervention led to statistically significant improvements for the comparison group, showing a reduction in both the number of visits and hospitalisations. As reflected in other studies, this could indicate that a multidimensional value-based intervention offers good value for money in terms of improving the frailty status of older people and reducing the costs of usual care [51]. However, this study was unable to find evidence against the hypothesis that a value-based intervention based on motivational interviewing and supported by digital solutions is effective in addressing frailty status with the current sample size. More data will be needed.

Among the strengths of this study, to date, no study has explored and addressed frailty from a holistic, personalised, and integrated value-based approach supported by digital solutions and motivational methodologies using a longitudinal design in older Spanish people living in the community. However, our findings must be interpreted within the limitations of this study, given that the paucity of previous similar studies with a sizeable sample makes it difficult to compare the present results with existing ones. While it is true that the present study showed no negative impact for participants, no statistically significant broad improvements have been observed, suggesting the need for further value-based multidimensional research in larger samples demonstrating efficacy in the specific management of frailty. Furthermore, despite the positive and significant relationship between frailty and loneliness, the present study did not reflect statistically significant improvements for loneliness, suggesting the need for future alternative screening and intervention programmes to prevent frailty and loneliness in people aged 65 years and older.

## 5. Conclusions

Given that frailty is a common condition in older people presenting multiple risks to their health and quality of life, it is important to note that the present study shows promising effects of implementing personalised and comprehensive value-based interventions supported by digital solutions, which may be a viable strategy to reverse this condition and improve patient outcomes. In conclusion, the results of the implementation of the ValueCare approach highlight the need to assess and address frailty from a multidimensional, comprehensive, and personalised value-based approach to reverse and curb this syndrome, considering that it is one of the most prevalent conditions with the greatest impact on the health of older people. To date, the identification, treatment, and prevention of frailty represents a challenge for health and social systems, due to its multidimensionality and the need to implement effective responses from a biopsychosocial approach and not only a unidimensional one based on attention to physical condition. From a practical point of view, this innovative value-based programme supported by digital solutions can be used as an effective alternative to other programmes (classical multicomponent exercise programmes and unidimensional programmes based on Fried’s physical phenotype) to optimise improvements in frailty syndrome, well-being, and quality of life in the older population. Finally, improving patient health outcomes in relation to the cost of care is a concern expressed by all stakeholders in the healthcare system, including providers, patients, researchers, and governmental organisations; therefore, more studies are needed that advocate the implementation of value-based care approaches with a large sample and that also consider follow-up beyond the end of the intervention period and thus can confirm and extend the findings of this study.

## Figures and Tables

**Table 1 healthcare-12-01754-t001:** Intragroup and intergroup differences in frailty.

		Intervention Group	Comparison Group	Intergroup Differences
TFI total score	Pre, Mean ± SD	4.91 ± 3.15	5.42 ± 3.25	*t* = 1.234; *p* = 0.109
	Post, Mean ± SD	5.17 ± 3.35	4.89 ± 3.11	*t* = −0.574; *p* = 0.283
	Intragroup differences *	*t* = −0.623; *p* = 0.267	*t* = −0.981; *p* = 0.164	
	Effect size **	0.32	0.24	
TFI physical score	Pre, Mean ± SD	2.34 ± 1.78	2.86 ± 2.16	*t* = 1.450; *p* = 0.074
	Post, Mean ± SD	2.72 ± 2.15	2.42 ± 1.98	*t* = −0.989; *p* = 0.162
	Intragroup differences *	*t* = −1.226; *p* = 0.112	*t* = 1.443; *p* = 0.076	
	Effect size **	0.14	0.17	
TFI psychological score	Pre, Mean ± SD	1.41 ± 1.11	1.37 ± 1.12	*t* = 0.602; *p* = 0.274
	Post, Mean ± SD	1.39 ± 1.08	1.37 ± 1.12	*t* = −0.289; *p* = 0.386
	Intragroup differences *	*t* = 0.072; *p* = 0.471	*t* = 0.129; *p* = 0.449	
	Effect size **	0.01	0.01	
TFI social score	Pre, Mean ± SD	1.11 ± 0.99	1.13 ± 0.94	*t* = 0.215; *p* = 0.415
	Post, Mean ± SD	1.05 ± 0.92	1.13 ± 0.92	*t* = 0.543; *p* = 0.294
	Intragroup differences *	*t* = 0.360; *p* = 0.360	*t* = 0.000; *p* = 0.500	
	Effect size **	0.06	0	

* Intragroup and intergroup differences were analysed using Student’s *t* test. ** Effect size was calculated using Cohen’s d (small effect = 0.2, medium effect = 0.5, large effect = 0.8).

**Table 2 healthcare-12-01754-t002:** Intragroup and intergroup differences in global health.

		Intervention Group	Comparison Group	Intergroup Differences
PROMIS Physical Health	Pre, Mean ± SD	42.07 ± 8.07	42.69 ± 9.06	*t* = 0.605; *p* = 0.273
	Post, Mean ± SD	43.82 ± 9.52	43.68 ± 8.83	*t* = −0.097; *p* = 0.462
	Intragroup differences *	*t* = −1.932; *p* = 0.029	*t* = −1.554; *p* = 0.062	
	Effect size **	0.22	0.15	
PROMIS Mental Health	Pre, n (%)	44.87 ± 6.90	44.69 ± 7.29	*t* = −0.255; *p* = 0.399
	Post, n (%)	45.44 ± 7.93	44.59 ± 8.08	*t* = −0.703; *p* = 0.242
	Intragroup differences *	*t* = −0.717; *p* = 0.238	*t* = 0.172; *p* = 0.432	
	% of change	0.08	0.02	

* Intragroup differences were analysed using McNemar’s test and intergroup analyses using chi-squared test. ** Effect size was calculated using Cohen’s d (small effect = 0.2, medium effect = 0.5, large effect = 0.8).

**Table 3 healthcare-12-01754-t003:** Intragroup and intergroup differences in loneliness.

		Intervention Group	Comparison Group	Intergroup Differences
UCLA scores	Pre, Mean ± SD	3.59 ± 1.27	3.69 ± 1.28	*t* = 0.306; *p* = 0.380
	Post, Mean ± SD	3.63 ± 1.19	3.69 ± 1.42	*t* = 0.729; *p* = 0.233
	Intragroup differences *	*t* = −0.196; *p* = 0.423	*t* = 0.000; *p* = 0.500	
	Effect size **	0.02	0	

* Intragroup differences were analysed using Student’s *t* test. ** Effect size was calculated using Cohen’s d (small effect = 0.2, medium effect = 0.5, large effect = 0.8).

**Table 4 healthcare-12-01754-t004:** Intragroup and intergroup differences of health-related quality of life.

		Intervention Group	Comparison Group	Intergroup Differences
5Q-5D-L5 general score	Pre, Mean ± SD	0.75 ± 0.23	0.72 ± 0.29	*t* = −1.791; *p* = 0.074
	Post, Mean ± SD	0.88 ± 0.16	0.83 ± 0.24	*t* = −1.434; *p* = 0.077
	Intragroup differences *	*t* = −5.215; *p* = 0.000	*t* = −4.695; *p* = 0.000	
	Effect size **	0.61	0.45	
5Q-5D-L5 mobility issues	Pre, n (%)	31 (25.4%)	35 (29.2%)	X^2^ = 0.430; *p* = 0.512
	Post, n (%)	17 (13.9%)	23 (19.2%)	X^2^ = 0.004; *p* = 0.949
	Intragroup differences *	McNemar; *p* = 1.000	McNemar; *p* = 0.152	
	% of change	−11.5%	−10.0%	
5Q-5D-L5 self-care issues	Pre, n (%)	12 (9.8%)	18 (15.0%)	X^2^ = 1.485; *p* = 0.223
	Post, n (%)	4 (3.3%)	15 (12.5%)	X^2^ = 4.161; *p* = 0.041
	Intragroup differences *	McNemar; *p* = 1.000	McNemar; *p* = 1.000	
	% of change	−6.5%	−2.5%	
5Q-5D-L5 pain issues	Pre, n (%)	77 (63.1%)	80 (66.7%)	X^2^ = 0.335; *p* = 0.563
	Post, n (%)	35 (28.7%)	43 (35.8%)	X^2^ = 0.213; *p* = 0.645
	Intragroup differences *	McNemar; *p* = 0.000	McNemar; *p* = 0.000	
	% of change	−34.4%	−30.9%	
5Q-5D-L5 daily activities issues	Pre, n (%)	27 (22.1%)	29 (24.2%)	X^2^ = 1.41; *p* = 0.707
	Post, n (%)	11 (9.0%)	18 (15.0%)	X^2^ = 0.355; *p* = 0.551
	Intragroup differences *	McNemar; *p* = 1.000	McNemar; *p* = 1.000	
	% of change	−13.1%	−9.2%	
5Q-5D-L5 anxiety depression issues	Pre, n (%)	50 (41.0%)	56 (46.7%)	X^2^ = 0.794; *p* = 0.373
	Post, n (%)	17 (13.9%)	34 (28.4%)	X^2^ = 2.697; *p* = 0.101
	Intragroup differences *	McNemar; *p* = 0.000	McNemar; *p* = 0.009	
	% of change	−27.1%	−18.3%	

* Intragroup differences were analysed using McNemar’s test and intergroup analyses using chi-squared test. ** Effect size was calculated using Cohen’s d (small effect = 0.2, medium effect = 0.5, large effect = 0.8).

**Table 5 healthcare-12-01754-t005:** Intragroup and intergroup differences in use of healthcare resources *.

		Intervention Group	Comparison Group	Intergroup Differences
Visits to primary care	Pre, Mean ± SD	4.22 ± 3.25	4.48 ± 3.77	*t* = 0.714; *p* = 0.238
	Post, Mean ± SD	3.37 ± 2.12	4.17 ± 3.17	*t* = 1.992; *p* = 0.024
	Intragroup differences *	*t* = 1.824; *p* = 0.036	*t* = 0.575; *p* = 0.283	
	Effect size **	0.22	0.06	
Hospitalisations	Pre, n (%)	17 (13.9%)	13 (10.8%)	X^2^ = 0.572; *p* = 0.450
	Post, n (%)	6 (4.9%)	11 (9.2%)	X^2^ = 0.448; *p* = 0.503
	Intragroup differences *	McNemar; *p* = 0.049	McNemar; *p* = 1.000	
	% of change	−9.0%	−1.6%	

* Intragroup differences were analysed using Student’s *t* test. ** Effect size was calculated using Cohen’s d (small effect = 0.2, medium effect = 0.5, large effect = 0.8).

## Data Availability

The data that support the findings of this study are available from the corresponding author upon reasonable request.

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
