# Peer review of "Efficacy of a Comprehensive and Personalised Approach for Frail Older People in Valencia (Spain): A Pre–Post Controlled Trial"

_healthcare, 2024, doi:10.3390/healthcare12171754_

Round 1

Reviewer 1 Report

Comments and Suggestions for Authors

The paper is well written and presents a clear argument to show the need for further preventative work addressing frailty.  Although others have made similar arguments, prevention/health promotion for frail older people is still lacking and so it seems there is need to continue to make this argument.  This paper goes one step further than calling for health promotion by providing evidence of a health promoting initiative.  The initiative discussed, however, had limited impact on the holistic health of frail older people.

General comments

No definition of what frailty actually is apart from lines 61-63 where it is stated ‘a dynamic state resulting from 61 deficits in any of the social, psychological, physical domains that contribute to health and 62 therefore requires identification, assessment and care of the condition from a biopsycho-63 social approach’ – I agree with this, however, I do think a clearer definition is needed alongside this. The Campbell and Buchner, 1997 definition may suffice here with some acknowledgement of cognitive frailty then followed by the statement quoted above. 

I also think that it is important that the authors acknowledge that frailty is a contested concept that can mean different things to different people and also that it is a term that carries much stigma.  I don’t think this requires a lot of additional work, just a couple of sentences to show acknowledgement of this important tension.  The following articles may be useful for this:

Cluley, V., Martin, G., Radnor, Z. and Banerjee, J., 2021. Talking about frailty: The relationship between precarity and the fourth age in older peoples' constructions of frailty. Journal of Aging Studies58, p.100951.

Cluley, V., Martin, G., Radnor, Z. and Banerjee, J., 2022. Talking about frailty: health professional perspectives and an ideological dilemma. Ageing & Society42(1), pp.204-222.

The intervention showed no impact on loneliness suggesting that an alternative approach is needed to combat loneliness, particularly as the authors make it very clear that loneliness has such a significant impact on propensity to frailty and its progression.  The authors do little to follow up on this.

I wonder if follow up/exit from the study has been considered i.e. how to continue to keep older people motivated to engage in health promoting activities.  I am unsure how long this initiative provides support for. Perhaps I missed this information but if it hasn’t been included, I think it is important to mention and also to provide information on how older people will be supported or not after the intervention to continue with health promoting behaviours.  I can see that tablets were available for use 6 months after the intervention but what happens after this? – are participants able to use their own device to continue to access support and if they don’t have their own device, is that the end?

There is no consideration of the impact of the use of digital technology.  While older people are increasingly familiar with this, the older people I work with frequently report that they do not want this sort of intervention and that they particularly don’t want to engage with chat bots.  I wonder how the use of digital technology was received by participants in this study?  How did older people feel about engaging in this way?

While the study showed a positive impact on number of hospital visits which could save money, the initiative showed limited benefits to the holistic health of frail older people leading the authors to conclude that more similar interventions are required on a larger scale.  I would question this logic and say perhaps an alternative approach is needed. The focus of this study is very much on personal responsibility in much the same way as the self-management and patient activation literature deals with personal responsibility.  I think there are inherent problems with this approach to prevention that could be the reason why the initiative had no impact on loneliness.  Personal responsibility is very much associated with a neo-liberal approach to healthcare.  I wonder if a more collective approach, encouraging group support may have been more effective.  I am not suggesting that this be included in the paper – these are just my personal thoughts. 

Author Response

General comment: The paper is well written and presents a clear argument to show the need for further preventative work addressing frailty.  Although others have made similar arguments, prevention/health promotion for frail older people is still lacking and so it seems there is need to continue to make this argument.  This paper goes one step further than calling for health promotion by providing evidence of a health promoting initiative.  The initiative discussed, however, had limited impact on the holistic health of frail older people.

General response: We would like to thank you for your words and the time you have taken to thoroughly review this manuscript.

Comment 1: No definition of what frailty actually is apart from lines 61-63 where it is stated ‘a dynamic state resulting from 61 deficits in any of the social, psychological, physical domains that contribute to health and 62 therefore requires identification, assessment and care of the condition from a biopsycho-63 social approach’ – I agree with this, however, I do think a clearer definition is needed alongside this. The Campbell and Buchner, 1997 definition may suffice here with some acknowledgement of cognitive frailty then followed by the statement quoted above.
I also think that it is important that the authors acknowledge that frailty is a contested concept that can mean different things to different people and also that it is a term that carries much stigma. I don’t think this requires a lot of additional work, just a couple of sentences to show acknowledgement of this important tension. The following articles may be useful for this:

Cluley, V., Martin, G., Radnor, Z. and Banerjee, J., 2021. Talking about frailty: The relationship between precarity and the fourth age in older peoples' constructions of frailty.
Journal of Aging Studies, 58, p.100951.

Cluley, V., Martin, G., Radnor, Z. and Banerjee, J., 2022. Talking about frailty: health professional perspectives and an ideological dilemma.
Ageing & Society, 42(1), pp.204-222.

Response: We really appreciate your comment and after careful consideration an additional paragraph has been included on page 2 (lines 55-63) reporting the controversy surrounding the term ‘frailty’ depending on the context in which it is used and the population, given the different perceptions between those who label (clinical setting) and those who ‘own the condition’ (non-clinical setting).

Also, given that the present study embraces frailty from a multidimensional definition, greater importance has been given to this conceptualisation by including on page 2 (Lines 70-72) an additional paragraph to facilitate a better understanding of this recent conceptualisation of frailty on which the study is based and which abandons the unidimensional character of the phenotypic conceptualisation of frailty previously mentioned in the text. It has been considered not to include other definitions in order to avoid distracting the reader as the main objective of the present study is to evaluate the effectiveness of a comprehensive, personalised and digitally supported programme on multidimensional frailty (including physical, psychological and social domains).

Comment 2: The intervention showed no impact on loneliness suggesting that an alternative approach is needed to combat loneliness, particularly as the authors make it very clear that loneliness has such a significant impact on propensity to frailty and its progression.  The authors do little to follow up on this.

Response: We thank you for your comment and fully understand what you are referring to, so we have included on page 11 (Lines 429-433) an additional paragraph that addresses the results about loneliness in this study as a limitation given the paucity of statistically significant improvement and the need to develop alternative approaches to detection and management that address the relationship between frailty and loneliness.

Comment 3: I wonder if follow up/exit from the study has been considered i.e. how to continue to keep older people motivated to engage in health promoting activities. I am unsure how long this initiative provides support for. Perhaps I missed this information but if it hasn’t been included, I think it is important to mention and also to provide information on how older people will be supported or not after the intervention to continue with health promoting behaviours. I can see that tablets were available for use 6 months after the intervention but what happens after this? – are participants able to use their own device to continue to access support and if they don’t have their own device, is that the end?

Response: We appreciate your comment and in view of the need to clarify the exit of participants from the study, an additional paragraph has been included on page 6 (Lines 269-278) giving more information on the actions that were taken to ensure a staggered exit from the study in order to alleviate a possible feeling of abandonment on the part of the participants, as well as to encourage them to continue their involvement in maintaining a healthy lifestyle.

 Comment 4: There is no consideration of the impact of the use of digital technology. While older people are increasingly familiar with this, the older people I work with frequently report that they do not want this sort of intervention and that they particularly don’t want to engage with chat bots. I wonder how the use of digital technology was received by participants in this study? How did older people feel about engaging in this way?

Response: Thank you very much for your comment. We fully understand your interest and therefore we would like to inform you that the following scientific publication ‘Implementation of a Comprehensive and Personalised Approach for Older People with Psychosocial Frailty in Valencia (Spain): Study Protocol for a Pre-Post Controlled Trial’ ( https://www.mdpi.com/1660-4601/21/6/715 ) includes information on the actions that were carried out to promote the acceptance and adherence of older people to the ValueCare application. Specifically, 2 co-design sessions were carried out with more than 200 participants in a pre-intervention phase in order to explore the opinions of the participants and to define the features and technical properties of the ValueCare solution.

Furthermore, an additional paragraph has been included on page 5 (lines 244-249) regarding the concern about the use of digital technologies in the older population and how this potential barrier was addressed in this study.

Comment 5: While the study showed a positive impact on number of hospital visits which could save money, the initiative showed limited benefits to the holistic health of frail older people leading the authors to conclude that more similar interventions are required on a larger scale.  I would question this logic and say perhaps an alternative approach is needed. The focus of this study is very much on personal responsibility in much the same way as the self-management and patient activation literature deals with personal responsibility.  I think there are inherent problems with this approach to prevention that could be the reason why the initiative had no impact on loneliness.  Personal responsibility is very much associated with a neo-liberal approach to healthcare.  I wonder if a more collective approach, encouraging group support may have been more effective.  I am not suggesting that this be included in the paper – these are just my personal thoughts. 

Response: We greatly appreciate these final reflections, which undoubtedly provide an interesting perspective to address the limitations of the present study and to guide future approaches.

Reviewer 2 Report

Comments and Suggestions for Authors

It was a pleasure reviewing this article. In my opinion, a key issue with this study is how the methods can be adapted to other settings, given the significant differences in social determinants across societies. Additionally, using the applications for elderly care faces challenges related to limited access to devices and low digital literacy. I hope my recommendations will make the author's article more complete.

1.      Abstract needs clarification. The number of participants, study settings (community, nursing home, hospital, etc.) must be specified. Who collected the data, when did the study take place? The current abstract only provides the idea of this study with lack of the important information.

2.      Introduction is too long. The first paragraph (from page 1 line 31 to page 2 line 50) can be removed and start with frailty incidence and impacts in Europe. What is “ICT” in this study? The objective of the study is unclear yet.

3.      Methods and results.

3.1  The authors mentioned “parallel-controlled” without any details of the control group. The most important information is how the authors manage the control group, and prevent the data contamination between control group and intervention group. For example, the participants in the control group may know someone who received intervention, and they may discuss something together. How did the researchers prevent it or manage it? This issue must be clarified.

3.2  The authors claim that The ValueCare project provide care for the people suffering from frailty, cognitive impairment, and/or multiple chronic disease (page 3 line 105) but the participants in this study were excluded one who had “cognitive impairment”. The statement you provided in the manuscript does contain a contradiction. Please clarify.

3.3  The inclusion criteria specify “aged 65 years or older” but the results said, “age ranging from 60 to 90 years old”. Please confirm.

3.4  Please explain how this number “71.9%” comes from? (page 4 line 151)

3.5  Strongly recommend providing “consort diagram” to clarify the participant enrollment in this study.

3.6  Please explain why not included the VAS in this study (page 4 line 183)

3.7  How the authors manage with the recall bias since the interview data about “over the past 12 months”

3.8  The information about ValueCare intervention is too subjective. Please provide more examples to help the readers understand how the study took place.

3.9  Please remove the effect size from all tables. I am not sure the importance of these numbers.

3.10          Since the intervention provides monthly discussion between general physicians and participants, that may be one of the reasons why participants in the intervention group showed less visits per year. However, all those data may be involved with recall bias and missing value. Please be aware of this concern.

4.      Discussion. Please add the limitation about the significant differences in social determinants across societies. Using the applications for elderly care faces challenges related to limited access to devices and low digital literacy.

5.      Did authors apply AI-assisted manuscript preparation? please clarify and state in the acknowledgement.

Author Response

General comment: It was a pleasure reviewing this article. In my opinion, a key issue with this study is how the methods can be adapted to other settings, given the significant differences in social determinants across societies. Additionally, using the applications for elderly care faces challenges related to limited access to devices and low digital literacy. I hope my recommendations will make the author's article more complete.

General response: We thank you very much for taking the time to review this article.

Comment 1: Abstract needs clarification. The number of participants, study settings (community, nursing home, hospital, etc.) must be specified. Who collected the data, when did the study take place? The current abstract only provides the idea of this study with lack of the important information.

            Response: We thank you for your comment and fully agree with the need for more specificity in the abstract. For this reason, additional information has been included (Lines 11-14) on the number and profile of study participants, the context in which the study took place and the time frame in which it took place.

Comment 2: Introduction is too long. The first paragraph (from page 1 line 31 to page 2 line 50) can be removed and start with frailty incidence and impacts in Europe. What is “ICT” in this study? The objective of the study is unclear yet.

Response: We greatly appreciate your suggestion, and therefore we inform you that the information from line 39 to 47 corresponding to references 8, 9 and 10 has been removed.  We have considered keeping the rest of the information since as the aim of the article is to address frailty in the older population we consider it appropriate to inform at the outset of the current state of population ageing and the challenges it poses for society and health systems where frailty is one of the main problems of ageing.

In addition, it is stated in the text, specifically in line 104, that ICT stands for ‘information and communication technology’. The use of the acronym ICT is necessary because one of the elements included in the ValueCare intervention is the use of a secure and robust digital solution, namely the ValueCare application, designed and used to monitor and support the ValueCare intervention (as stated in line 206 and in the paragraph comprising lines 251-268). In addition, another aspect to note in relation to the term ICT is that all participants in the intervention group received a digital device, specifically a tablet through which to use the ValueCare app.

Comment 3: The authors mentioned “parallel-controlled” without any details of the control group. The most important information is how the authors manage the control group, and prevent the data contamination between control group and intervention group. For example, the participants in the control group may know someone who received intervention, and they may discuss something together. How did the researchers prevent it or manage it? This issue must be clarified.

            Response: Thank you for mentioning this relevant aspect. Participants in the control group received standard care from their healthcare center, while those in the intervention group received personalized care. Although participants from both groups may have shared information, the intervention plan was tailored to each individual, making the goals and methods specific to each participant and not necessarily applicable to others. Additionally, participants in the intervention group had access to the ValueCare app, which was not available to the control group, and this app was one of the most relevant means through which they received the intervention.

Comment 4: The authors claim that The ValueCare project provide care for the people suffering from frailty, cognitive impairment, and/or multiple chronic disease (page 3 line 105) but the participants in this study were excluded one who had “cognitive impairment”. The statement you provided in the manuscript does contain a contradiction. Please clarify.

            Response: Thank you very much for your comment. We would like to clarify that line 105 (after modifications line 114) refers to the ValueCare project in general, which has been implemented in a total of 7 pilots in 7 different European cities covering different chronic diseases, cognitive impairment or frailty. However, in the pilot in Valencia (Spain) in which the present study is framed, frailty has been the condition addressed.  

In addition, to clarify any possible doubts, an additional paragraph has been included on page 3 (Line 117-123) which lists the different pilot sites where the project has been implemented and specifies that each one has focused on a specific condition (chronic diseases, cognitive impairment or frailty).

 Comment 5: The inclusion criteria specify “aged 65 years or older” but the results said, “age ranging from 60 to 90 years old”. Please confirm.

            Response: Thank you very much for your appreciation. The age in the results (line 298) has been changed to 65 years old. It can now be read as follows: ‘age ranging from 65 to 90 years old’.

Comment 6: Please explain how this number “71.9%” comes from? (page 4 line 151)

            Response: Thank you for your comment. The percentage ‘71.9%’ refers to the proportion of final individuals who completed the ValueCare intervention.

Comment 7: Strongly recommend providing “consort diagram” to clarify the participant enrollment in this study.

            Response: We greatly appreciate your suggestion. We consider this to be very important and would therefore like to let you know that the Diagram detailing the flow of clinical trial participants from recruitment to the last follow-up contact for control and intervention subjects is available in a previous publication based on the pre-post controlled trial study protocol presented here. Please feel free to visit the publication for full details: https://www.mdpi.com/1660-4601/21/6/715

Comment 8: Please explain why not included the VAS in this study (page 4 line 183).

            Response: We acknowledge that the VAS offers valuable information, but the ValueCare research project decided not to include it due to the length of the questionnaire.

Comment 9: How the authors manage with the recall bias since the interview data about “over the past 12 months”.

            Response: Thank you for bringing up this concern. We aimed to minimize recall bias by pilot testing the questionnaire to identify and revise questions that were particularly susceptible to it. During the interviews or questionnaires, participants were supported in recalling events more accurately without being led.

 Comment 10: The information about ValueCare intervention is too subjective. Please provide more examples to help the readers understand how the study took place.

            Response: Thank you very much for your comment. We would like to clarify that the ValueCare intervention has at all times been based on existing scientific evidence and has involved all stakeholders from the beginning of the project, both in the definition of the value concept and in the definition and design of the features and properties of the ValueCare application.  In addition, data collection and measurement were conducted using an assessment questionnaire that included the International Consortium for Health Outcomes Measurement (ICHOMs) thus guaranteeing the objectivity of the results.  Also, we would like to inform you that in the line 148 (page 3) readers are invited to visit the published Study protocol for any details on the method, design and procedure of this trial. 

Comment 11: Please remove the effect size from all tables. I am not sure the importance of these numbers.

            Response: To our understanding, effect size is important because it provides a measure of the magnitude of the intervention's impact, beyond just statistical significance. Therefore, if it's not inconvenient, we would prefer to include it in the tables.

Comment 12: Since the intervention provides monthly discussion between general physicians and participants, that may be one of the reasons why participants in the intervention group showed less visits per year. However, all those data may be involved with recall bias and missing value. Please be aware of this concern.

            Response: Thank you for bringing up this concern. The monthly discussions were limited to topics directly related to the research and to our knowledge did not include any additional consultations.

Comment 13: Discussion. Please add the limitation about the significant differences in social determinants across societies. Using the applications for elderly care faces challenges related to limited access to devices and low digital literacy.

            Response:  Thank you very much for your suggestion. We fully understand your interest and would like to share that the scientific publication titled ‘Implementation of a Comprehensive and Personalised Approach for Older People with Psychosocial Frailty in Valencia (Spain): Study Protocol for a Pre-Post Controlled Trial’ provides detailed information on the actions taken to encourage the acceptance and adherence of older individuals to the ValueCare application. In particular, two co-design sessions were conducted during the pre-intervention phase with over 200 participants to gather their feedback and define the features and technical specifications of the ValueCare solution. Moreover, on page 5 (lines 244-249), we included a paragraph addressing concerns about the use of digital technologies among the older population and how this potential barrier was addressed in our study.

We would also like to inform you that the project provided electronic devices to all participants of the Intervention group. In the case of the Valencia pilot, participants received free tablets with internet access available for up to 6 months after the end of the intervention, thus overcoming the possible financial constraints faced by the participants.

Comment 14: Did authors apply AI-assisted manuscript preparation? please clarify and state in the acknowledgement.

            Response: Thank you very much for your comment. We did not use AI to support the production of this manuscript.

Round 2

Reviewer 2 Report

Comments and Suggestions for Authors

Thank you to the author for answering all questions thoroughly and patiently. The article is now of sufficient quality for publication.